# Developing an Internet-Based Trauma Recovery Nursing Intervention Based on Swanson’s Theory of Caring for Trauma Recovery

**DOI:** 10.3390/ijerph18136715

**Published:** 2021-06-22

**Authors:** Sunah Kim, Go-Un Kim, Wongyeong Lee, Jinyoung Park

**Affiliations:** 1College of Nursing, Yonsei University, Seoul 03722, Korea; psy0962@yuhs.ac (S.K.); kgudfc@naver.com (G.-U.K.); DAISY2WK@yuhs.ac (W.L.); 2Mo-Im Kim Nursing Research Institute, Yonsei University, Seoul 03722, Korea

**Keywords:** nurses, nursing, post-traumatic stress disorder, internet-based nursing intervention, Swanson’s theory of caring

## Abstract

This study aimed to develop an Internet-based psychiatric nursing intervention, based on Swanson’s theory of caring, that could alleviate post-traumatic stress symptoms and improve the health functioning of adults who had experienced trauma. To evaluate the appropriateness and understanding of the program content, expert advice and content validity were examined using the content validity index. To analyze the participants’ responses and evaluate the appropriateness of conducting Internet-based programs, an online intervention program was implemented with five nurses between February and April 2020. The trauma recovery nursing intervention program was for adults who had experienced trauma and consisted of eight sessions, each lasting 30 min. It included a spoken audio track that was accompanied by visual prompts to provide knowledge on traumatic stress and introduce trauma recovery methods. Its online format made it convenient and widely accessible, and allowed for a low-intensity, cost-effective educational program. The program focused on providing self-help guidelines for clients to develop their mental and physical well-being. The results provide evidence that mental health nurses could develop and implement online programs to care for people who have experienced trauma. Such programs could provide trauma recovery for marginalized people unable to otherwise receive help.

## 1. Introduction

Psychological trauma refers to mental damage that occurs when one’s ability to overcome or integrate is exceeded as the result of a stress-inducing event [1]. This is a widely occurring phenomenon that affects people regardless of their region, race, educational level, or socioeconomic status. Psychological trauma is deeply connected to the causes of post-traumatic stress disorder (PTSD), anxiety disorder, sleep disorder, behavior disorder, depression, and suicide [1,2]. Trauma is both a mental health problem and a serious social problem; however, it can be alleviated through prevention and early treatment [1,2]. However, if treatment and care period are delayed, trauma can progress into a chronic disease and result in a deterioration in quality of life amid the effects of PTSD [1,3]; thus, a national-level counter strategy must be developed [4,5].

Among the noninvasive social psychological treatments that are available for trauma patients, cognitive-behavioral therapy has been confirmed as the most effective intervention [6]. However, immediately after experiencing trauma, many patients with high anxiety struggle to focus during cognitive-behavioral therapy and do not experience relief from their physical symptoms [7]. Studies have confirmed that exposure therapy is also effective in treating patients with chronic post-traumatic stress symptoms; however, it is a difficult strategy to apply when patients exhibit multiple physical symptoms. Moreover, the lack of trained professional therapists limits nurses’ ability to learn the treatment [8]. Recently, studies reported that eye movement desensitization and reprocessing (EMDR) has a therapeutic effect in people who have experienced trauma; however, this technique also has high training costs. Additionally, certification requires thoroughly supervised training at the EMDR Society headquarters in the United States and general nurses lack access to the certification programs. Thus, its use is limited [8]. As such, there is a need for accessible and effective options for the treatment of post-traumatic symptoms. Therefore, the aim of this study was to develop an Internet-based self-help program for adults who have experienced a traumatic event.

### 1.1. Background

In various countries, professionals from health-care specialized fields are participating in the respective “Trauma Health” fields within their communities and are striving toward competency development [2,9]. In particular, therapists who care for people experiencing trauma can easily experience burnout; therefore, there is a great need for training programs within this profession [10]. For this reason, a national-level response strategy that provided a significant amount of support for building a network of experts and the providing human resource training was developed [2,10].

In Korea, mental health experts (e.g., doctors, nurses, social workers) are mobilized at the “Community Mental Health Welfare Center” [4]. Mental health experts perform monitoring, evaluation, crisis counseling, and case management necessary for psychological recovery for those who have experienced traumatic events, and vulnerable people in their families and communities over a long period of time immediately after trauma occurrence [3].

Trauma recovery requires an integrated approach, from individual psychological and emotional problems to changes in social structure and environment [3]. However, the lack of an integrated and systematic education program for mental health nurses at the national level has been shown to make trauma therapy difficult, which further promotes psychological burnout of mental health professionals [4,5].

Many psychiatric nurses still struggle with deviating from the perspective of psychiatric hospital-oriented trauma treatment. Furthermore, decreased interest in the nursing field has led to a serious lack of nursing personnel who have received systematic trauma treatment training [4,5]. However, in Korea, although social welfare and clinical psychology college majors actively participate in strengthening their capacity for trauma management and treatment, there are few opportunities to engage in trauma treatment training in the nursing field [5]. Nurses’ lack of competency in trauma intervention is thought to be a contributing factor towards the lack of participation by nurses [11].

Previous studies have found that resilience and the ability to cope with stress buffer the negative symptoms associated with trauma experiences and contribute to improving mental health [12,13]. In addition, guided self-help nursing interventions can assist in the management of various psychiatric conditions, such as depression and anxiety, in patients with PTSD, and help these patients find satisfaction and happiness in life while gaining a deeper understanding of themselves [14]. Nursing interventions are based on a holistic understanding of human beings; however, interventions that can integrate physical, mental, social, and behavioral interventions within effective therapeutic strategies must still be developed [15].

This study used Swanson’s theory of caring [16], a middle-range theory, as its theoretical basis. Swanson’s theory of caring was initially developed for parents who experienced fetal miscarriages; however, it has been used in various situations, including for parents of premature babies, and has been used as the theoretical basis for several previous studies [17]. In addition, researchers have reported that level of care and patient safety improved when Swanson’s theory of caring was used as the theoretical basis for supporting nursing practice in several hospitals in the United States, as did the level of satisfaction in the care provided to patients by nurses [18]. Swanson’s theory of caring suggests that “client well-being” can be maintained while executing a five-step process: “maintaining belief,” “knowing,” “being with,” “doing for,” and “enabling” [19]. This process is applicable to both those who have experienced trauma and for the nursing staff.

People often do not receive early professional treatment following traumatic experiences because they are not aware of the negative impact that trauma can have on their mental health. Moreover, patients can present with a wide variety of symptoms, including physical symptoms and sleep disturbances [20]. Patients may be concerned that they could re-experience trauma during the course of the treatment, and may struggle to connect with the appropriate therapist [2,21,22]. Additionally, face-to-face treatment is time-consuming and expensive, and appropriate services are difficult to provide in isolated places, such as rural areas [14,23].

When it comes to overcoming these limitations, Internet-based interventions are promising treatment options compared to face-to-face interventions. According to Lewis et al. [14], Internet-based interventions require much less time from therapists and clients, and guided psychological interventions are a treatment option that could help broaden access to effective interventions. In their study, the Internet-based intervention group reported significantly lower PTSD, depression, anxiety, and dysfunction scores than the face-to-face intervention group. Based on such previous research, this study aimed to contribute to ease of management and low-cost operation by utilizing popular existing Internet media such as e-mail, video calls, and video lectures.

### 1.2. Aim

The aim of this study was to develop an Internet-based psychiatric nursing intervention that could alleviate post-traumatic stress symptoms and improve the health functioning of adults who have experienced trauma.

## 2. Materials and Methods

### 2.1. Design

This was a methodological research study that developed an Internet-based program based on Swanson’s theory of caring [16]. The process involved developing the program’s content, and then evaluating its Internet-based implementation. The evaluation was performed using five nurses as a convenience sample who role played as clients for the purposes of developing this intervention. The nurses worked in nonmental health-related areas and had more than three years of nursing experience. The five participating nurses assessed the appropriateness of the program’s online configuration and provided feedback on their experience with the program. The researchers conducted the program in the role of the therapist. The program was run individually online.

### 2.2. Ethical Considerations

Approval for this study was obtained from the institutional review board of Yonsei University (Y-2019-0083). The participants expressed their intention to voluntarily participate through the recruitment announcement and were provided with a description of the study. This material explained the study’s purpose, potential risks, and benefits. Participants were informed that participation was voluntary and they could withdraw from the study at any time.

### 2.3. Conceptual Framework

Swanson’s theory of caring is based on the experiences of women who had undergone abortions or given birth to a premature baby [16]. The theory emphasizes that the focus of nursing is not on human illnesses but on helping patients achieve, maintain, and recover optimal levels of well-being [18]. In addition, as a middle-range theory, it provides the basis for nursing practice and applies each caring process to nursing. This theory describes five caring processes: “maintaining belief,” “being with,” “knowing,” “doing for,” and “enabling” (Figure 1).

The concept of “maintaining belief” emphasizes the trust between the client and the nurse. To maintain hope and recovery, clients should help and respect themselves [16,24]. In this study, the research team used response letters, which aided in developing a strong and deep trust between the therapist and client, that were related to “maintaining belief” from the first to the last session. Therapists aimed to provide an understanding and supportive attitude, as well as hope, in the response letters.

In the “being with” stage, the nurse reports on the necessity behind recognizing the client’s feelings by allowing them space to share their feelings and experiences. One study reported that nurses should communicate clearly with clients and not feel burdened [16]. In this study, the “being with” aspect of the interaction between the client and therapist was also applied to the response letter through interactive communication. From the response letter, clients could recognize that the therapist was listening to their experiences and feelings. When clients who had experienced trauma were treated by nurses, they found that they were able to accept and respect their existence.

The “knowing” stage aims to develop the client’s understanding of the meaning of the events that have occurred in their lives. From this perspective, the nurse and client try to create knowing and understanding around the human responses to recovery, rather than emphasizing or avoiding traumatic experiences [16,17]. In this study, “knowing” focused on getting to know and respect the clients themselves: grasping experiences, thoughts, and feelings, and grasping their mutual relationships.

The “doing” stage is characterized by the nurse caring for the client. One study reported that nurses should help maintain client well-being by allowing the nurses to protect and care for clients and by allowing clients to set goals for themselves [24]. In this study, “doing” emphasized the client’s self-help process by recognizing their traumatic experiences, maintaining rational recognition in stressful situations, and controlling negative emotions.

In the “enabling” stage, nurses allow clients to change their daily routine and solve problems on their own as they encounter new events [16,19]. In this study, the concept of “enabling” provided the perspective that clients could solve their problems and find meaning in life. To find gratefulness and satisfaction in daily life, clients learned effective communication skills and stress management strategies.

### 2.4. Program Development Process

#### 2.4.1. Stage I. Program Content Development Process

Based on the theoretical framework and literature review, the research team set the goals, theme, and main content of the intervention program, and determined the duration of the eight-session program. The qualification requirements for the researchers who conducted the program were identified.

To confirm the content for each session, the content that we developed was reviewed, revised, and supplemented by experts on the program draft in paper documents. The experts comprised one trauma counselor at a counseling center and six mental health nurses at mental health services and mental health rehabilitation centers. Experts assessed the effectiveness of each session’s purpose and content using a four-point Likert scale, where 1 = completely inappropriate, 2 = not appropriate, 3 = appropriate, and 4 = very appropriate. The content validity index (CVI) was also calculated.

To evaluate the appropriateness and understanding of the program content, expert advice and the CVI were checked.

The suitability of the overall purpose of the program and the goals for each session were verified. The CVI was 3.2 points (86.1%), thus verifying the content validity of the program. Moreover, the program was modified and supplemented based on the expert advice.

#### 2.4.2. Stage II. Internet-Based Program Execution Process Development Process

Through evaluation of the adequacy of the online program, we analyzed the participants’ responses to the Internet-based eight-session program. We confirmed the corrections and supplementations with the participants to establish their understanding of the composition and content of the program, the difficulty of the vocabulary, and the adequacy of the time provided. The final program was developed by applying the feedback that was received. Table 1 shows the feedback from the participants and the revisions that were made that reflected their feedback. During the evaluation of the feedback, the research team recognized that the participants wanted more active interaction with the therapists. Therefore, the research team developed a response letter for each session. Following the correction and supplementation of the content, and the online application of the program using the method developed from participants’ responses in the evaluation, the program was finalized.

## 3. Results

We developed a trauma recovery nursing intervention program for adults who have experienced trauma. This Internet-based program consisted of eight 30 min sessions. The program included a spoken-audio track that was accompanied by visual prompts. It aimed to provide knowledge on traumatic stress and introduce methods to recover from trauma. We chose an online format to make it convenient and widely accessible, and to allow for a low-intensity, cost-effective educational program. The theoretical framework for this study was based on Swanson’s theory of caring [16] and a literature review of trauma interventions. The specific topics covered in the training program are outlined in Table 2. The detailed content of each session is described below.

### The Eight-Session Program

First, before the start of the main program, the background and purpose were explained to participants and an overall introduction on how the program would be conducted was provided.

Session 1 emphasized self-understanding and self-respect. Clients had time to understand themselves by remembering things that were either sad or happy and recording them. Clients were provided with information on PTSD, and we confirmed that the clients were not experiencing PTSD without support. In addition, clients were encouraged to determine their undiscovered strengths. They were also encouraged to consider how they could use their strengths in their lives. After Session 1, clients practiced abdominal breathing to calm their minds, which they may have been struggling with due to the task of recalling painful memories.

Session 2 helped clients be more vulnerable than in the first session. Clients thought of people who had hurt them in their lives and faced the negative emotions they felt. To turn their negative emotions into positive ones, they were reminded of the aspects of their life for which they could be grateful. The clients were given information on flashbacks and how to cope with them if they experienced any. After Session 2, the clients practiced gradual muscle relaxation, a type of body stabilization. Body stabilization focuses on physical and emotional sensations, and on the present rather than the past.

Session 3 focused on clients accepting and sharing trauma experiences on their own. Through self-exposure, clients were encouraged to understand and share how the difficult situations that were addressed in the second session affected their feelings, thoughts, and behavior. It helped evoke and discharge negative emotions and thoughts. In addition, clients were trained to control their emotions by turning negative thoughts into positive ones. After Session 3, clients practiced using their imagination. Imagination training calms emotions through the repeated, intermittent flashing of negative events or emotions on a computer screen.

Session 4 highlighted the benefits of maintaining a rational viewpoint in stressful situations. Clients could look at their situation objectively and identified the benefits of rational thinking. Through examples of irrational thoughts (e.g., black and white logic, exaggeration/reduction, mind reading), they recognized that they were often irrational themselves. Clients learned that, through a rational worldview and perspective, they could have positive emotions. After Session 4, body stabilization exercise was performed.

Session 5 emphasized the management of the negative emotions felt by clients who experienced trauma. Clients checked the emotions they often felt to determine whether they experienced positive or negative emotions more frequently. Clients learned the benefits of positive emotions and were trained to distinguish between the thoughts and emotions experienced in certain situations through examples. They were also trained to change their emotions and behaviors by reminding themselves of stressful situations and transforming the emotions and thoughts that they were experiencing at that time into rational thinking. After completing the Session 5, the abdominal breathing exercises were repeated.

Session 6 focused on developing effective communication skills. Clients learned the communication skills required to their express emotions and thoughts and change their behavior. After identifying their communication types and habits, clients shared their difficulties and conflicts related to communication with their therapists via response letters. They also learned to express negative emotions or deal with negative situations using “I-message” language. Through the use of examples of inefficient communication, clients became committed to avoiding inefficient communication. After Session 6, body stabilization exercise was repeated.

Session 7 focused on managing stress and leading a healthy lifestyle. Clients were asked to learn four strategies that would assist with stress management: cognitive control, behavior control, physiological relaxation, and daily life management. Clients identified a situation and planned strategies to reduce stress. In addition, the application of a stress response strategy through identification of automatic thinking and adjustment to alternative and rational thinking was suggested. Finally, clients learned the “butterfly embrace” for physiological relaxation.

In Session 8, clients were empowered to find meaning in their everyday lives. They recognized that they could enjoy a peaceful life by solidifying their strengths and that they could find happiness and meaning in their own lives. Through summarizing the information in Session 1–7, clients could confirm what had changed recently in their daily lives. They were provided with a perspective of gratitude and satisfaction in daily life. Through integration of the “knowing,” “doing for,” and “enabling” topics, clients who had difficult experiences could recover from trauma. At the end of the Session 8, the body stabilization exercise was repeated.

During each of the aforementioned program sessions, the research team sent individual response letters to the clients. The response letter was created based on the evaluation feedback. To ensure mutual communication between the therapist and client, the therapist interacted with the client continuously and fulfilled the “maintaining belief” and “being with” domains for the client. Response letters were provided at the end of each session. Based on the content of the client’s responses throughout the program, we communicated with clients individually to provide support and hope through positive feedback. In addition, we motivated clients to continue participating in the program without giving up. The standardized response letter provided a framework for therapists and we expected that it would improve the service quality. In addition, we provided a guide in the responses to address each client’s individual circumstances or feelings.

## 4. Discussion

This study developed an Internet-based psychiatric nursing intervention for trauma recovery based on Swanson’s theory of caring [16], a middle-range theory that can be applied directly to nursing practice in the mental health field. Swanson’s theory of caring provides a conceptual framework for a therapeutic approach to those suffering from trauma. The program that was developed was constructed based on five key concepts that ultimately aimed to maintain the client’s state of well-being: “maintaining belief”, “knowing”, “being with”, “doing for”, and “enabling”. The concepts were not mutually exclusive, and each concept was incorporated into each session. Thus, they acted as an overlapping process and influenced and interacted with each other. Swanson’s theory of caring is considered both incredibly relevant and applicable to the field of nursing practice with respect to the views that it presents of humans and the relationship between therapist and client [24,25]. The program that was developed in this study did not provide symptom-oriented interventions that only focused on traumatic events or experiences from the therapist’s professional standpoint. We focused on providing a self-help guideline for the clients that would allow them to develop their mental and physical well-being.

The participants were anonymous, and the researchers promised that their information would never be leaked in the study. In each session, the participants were asked several questions on the difficulties that they experienced. The researchers provided individual feedback on their answers using response letters. A previous study reported that nursing interventions that were provided based on the theory of caring for individuals who had experienced intimate partner violence had a therapeutic effect [9]. As the participants of that study had recently experienced difficulty with self-exposure and needed to develop their beliefs with the nurses, the nurses continued to maintain the patient’s belief throughout the nursing intervention [9].

For the “maintaining belief” and “being with” domains, response letters were sent to the clients individually after each session. The response letters were developed at the request of the participants during the evaluation. The participants proposed that interaction and support with the therapist would be effective. The response letter was created to facilitate interactive communication with the client and clients were provided feedback on their answers to the questions. The “being with” domain emphasizes the intimate relationship with the client as the center of care [24]; that is, it is based on the client’s bonding and attachment [26]. Using themselves as therapeutic tools, nurses must be emotionally open to client reactions and be able to reassure clients with a warm attitude [26,27]. The reasons for failure in psychiatric counseling have been reported to include clients’ lack of trust, intimacy, and motivation in relation to the therapist [28]. In this study, after the end of each session, based on the client’s responses, they were sent a supportive and positive response letter to give them a sense of intimacy and hope. Moreover, the response letters played a role in motivating and encouraging clients to continue their participation in the program without giving up.

In the Session 1 and 2, for the “knowing” dimension, clients were helped to understand their current state of mind, past traumatic experiences, moods, thoughts, and strengths. Specifically, clients were made aware of their strengths and ability to overcome the effects of trauma. It is imperative that nurses understand and respect clients from a holistic perspective. The patient can overcome a crisis by getting to know and respecting themselves through the process of knowing [24]. The sessions helped clients to become familiar with and respect themselves through knowledge of their strengths. Finding strengths is a positive psychological intervention that can relieve depression and lead to improved well-being. Its effectiveness has been demonstrated in several studies [29,30]. Clients increase their happiness in life by finding and enhancing their strengths.

In Sessions 3−5, the concept of “doing for” focused on enhancing the clients’ competencies. Clients in this program could accept and share their traumatic experiences through self-disclosure. To do this, clients were asked multiple questions through noninvasive techniques. Moreover, they were provided with a wealth of knowledge on trauma that enabled them to strengthen their own capabilities through constant training. The characteristics of competence included knowledge, ongoing adjustment to learning collective skills, and self-confidence [26,31]. This program provided a self-help guide for clients. It allowed them to accept trauma-related experiences by themselves and develop a rational perspective while controlling negative emotions. Previous studies reported that Internet-based, self-help interventions for patients with PTSD and eating disorders were less expensive than face-to-face treatment [14,32]; however, they were effective in reducing negative emotions such as depression and alleviating its symptoms. In particular, they may have had a better effect during early treatment [32].

For the “enabling” dimension, the client’s changes were identified, and clients were encouraged to continue to take care of themselves through intrinsic motivation in Sessions 6−8. “Enabling” emphasizes the need for continuous validation and confirms the appropriate results of caring [24,26]. Engagement, meaning, positive relationships, and achievement-based interventions are effective strategies for increasing well-being and improving depressive symptoms [29]. Clients were encouraged to feel a sense of accomplishment for the changes they had made and to praise and reward themselves. In addition, based on what they had gained through this program, they were encouraged to practice appropriate communication in negative relationships and situations in their daily lives and continue their stress management strategies. Thus, the importance of finding meaning in their daily lives and maintaining a sense of happiness was emphasized.

This study was also significant because it was an online nursing intervention program. Although online treatment and intervention programs are provided in various clinical fields, there are few online nursing interventions in the mental health nursing field [3,23,33]. The greatest advantage of online programs is the lack of time and space constraints. Face-to-face counseling has become increasingly difficult since the onset of the coronavirus disease 2019 pandemic [21]. In place of face-to-face counseling, Internet-based intervention programs are expected to help greater numbers of people who have experienced trauma. Online intervention programs have the advantage of being more cost effective than face-to-face consultations [34,35]. They also allow for 1:1 interaction and can provide interventions through two-way communication [3,33].

Often in psychiatric counseling cases, patients avoid counseling due to the fear of stigma [22]. The stigma associated with mental health problems such as traumatic stress can act as a barrier to mental health treatment and positive outcomes [2]. Online programs are advantageous as they reduce the stigma and the burden of face-to-face counseling, allowing clients to be completely vulnerable and face their own problems.

Although Internet-based therapies have posed technical difficulties [23,33], the development of such interventions is incredibly advantageous because it contributes to the spread of standardization. The trauma recovery intervention program that was developed in this study presented a standardized framework and material for eight sessions. We expect that a standardized intervention program will improve the quality of nursing intervention services and enhance the nursing intervention competency [18]. One study that provided education programs to nurses based on Swanson’s theory of caring found that nurses who were trained according to the caring framework had much higher levels of care than those who were not [18]. Moreover, both patients and nurses expressed greater satisfaction due to the reinforcing of the relationship between nurses and patients [35]. We also expect that this program will enhance satisfaction not only for clients who have experienced trauma but also for the nurses who provide nursing care.

### Limitations

A limitation of the study is that it was not possible to evaluate the effectiveness of the program. Another limitation is the small number of participants.

## 5. Conclusions

The trauma recovery program that was developed in this study was based on Swanson’s theory of caring [16]. Caring has long been recognized as the center of nursing care and is increasingly becoming a key concept [24]. Therefore, this study is of great significance as it provides evidence that mental health nurses could develop and implement programs to care for people who have experienced trauma. The program will help to alleviate PTSD symptoms and, ultimately, allow for patients to live happy and meaningful lives. An online program could even provide nursing interventions for trauma recovery for marginalized people who are unable to receive help from nurses for physical reasons, such as distance and time. In addition, as one of the advantages of online programs is the characteristically swift dissemination of information, we propose that nurses in various regions could easily learn from trauma intervention programs and provide high-quality nursing interventions to their clients. The program will need to be administered to clients who have experienced trauma, and continued research will need to confirm whether they experience symptom relief and for how long. In addition, it will be important to widely disseminate the program, educate nurses on trauma and use of the program, and investigate whether there is an increase in nurses’ trauma intervention competency as a result.

## Figures and Tables

**Figure 1 ijerph-18-06715-f001:**
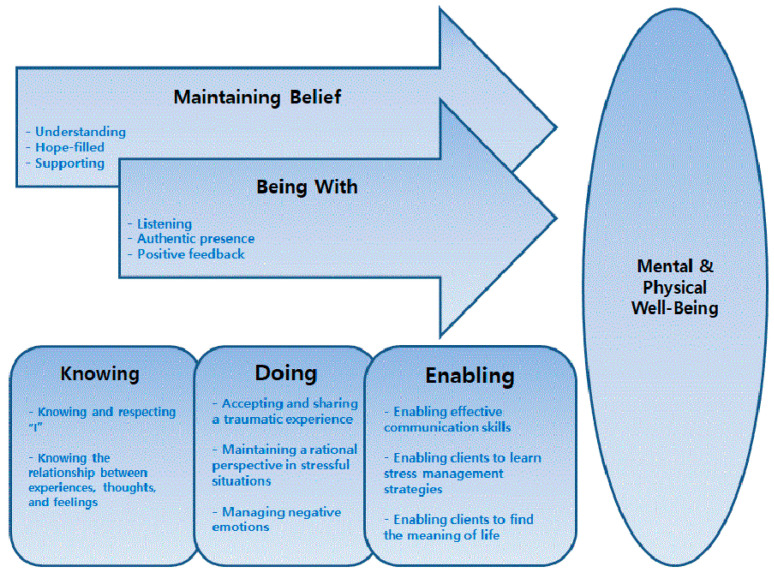
Conceptual framework for the program.

**Table 1 ijerph-18-06715-t001:** Participant feedback and revisions to the eight-session Internet-based program.

Participant Feedback	Modifications Reflecting the Feedback
Lack of two-way communication *“There is a lack of intimacy and communication with the therapist.”* *“There was no communication with the therapist. It felt as though the program was delivered unilaterally.”* *“It was unfortunate that I could not build* *a* *rapport with therapist.”* *“It would be good if a support system could be formed through constant communication with the therapist.”*	Added individual response letter for interactive communication. If the client wished, an on/offline interview was made possible at any time.
Difficulty with narrative responses to questions *“It was difficult because I was unfamiliar with describing narrative responses.”* *“I was hesitant about what to write for the question.”* *“It would be better to reduce the number of narrative questions and provide brief options for the answers.”*	A guideline for the response was provided with examples of possible narrative responses.
Lack of program instruction *“It would be nice if there were a guide available for the program before starting it.”* *“I started without knowing what kind of program it was, but it took more time than expected and it was difficult.”*	Before the start of the program, detailed explanations were provided on the background and purpose of the program, and on how to respond.
Need for specific questions *“Be specific about what you are asking. I wasn’t sure if I was responding correctly when I responded.”* *“When answering the question, it seemed that a* *specific* *example* *for* *each question is needed.”* *“It would be nice if you could explain exactly what the question was asking. For example, I was confused about how much anxiety ‘anxiety’ meant.”*	The questions in each session were specifically changed and examples of the response were provided to make it easier for the clients to answer.
The amount and structure of the content in relation to providing it online *“The content was only composed of text, so I grew tired easily.”* *“It would be nice to make the program less boring by including things like pictures and videos.”* *“It was difficult to concentrate on the program because there were too many explanations provided in text on my cellphone.”*	The program was amended through reduction and simplification of the written content, and through the addition of pictures, photos, and videos that could be provided online.

**Table 2 ijerph-18-06715-t002:** The eight-session program.

Session	Swanson’s Caring Theory Process	Themes and Major Content	Guided Self-Help Questions for Trauma Recovery
1	Knowing	Knowing and respecting “I” Learning knowledge related to trauma Understanding the importance of living happily Trying to find my strengths Practicing mental control through abdominal breathing	Asking questions to find strengths Asking questions about happy experiences and situations Trying to introduce experiences of difficult times
2	Knowing	Knowing the relationship between experiences, thoughts, and feelings Recognizing changes after painful experiences Finding meaning and goals in life to find happiness away from traumatic experiences Understanding the link between physical activity and mental health Learning body stabilization Practicing mental control through body stabilization	Asking questions about difficult situations and hard times Asking questions about how to cope with getting out of difficult situations Remembering positive emotions and gratitude
3	Doing for	Accepting and sharing traumatic experiences Using self-disclosure Identifying the influencing factors of negative emotions and thoughts Learning to manage negative emotions Practicing mental control through abdominal breathing	Sharing traumatic experiences Remembering positive thoughts and emotions in daily life Asking questions that establish the relationship between negative thoughts, emotions, and behaviors Changing negative emotions using the “imagination”
4	Doing for	Maintaining a rational perspective in stressful situations Trying to have objectiveness Identifying the benefits of rational thinking Expanding one’s worldview and perspective in a safe environment Practicing mental through body stabilization	Remembering and enjoying strengths Recognizing the degree of negative emotion Applying a management strategy to change negative emotions Writing from a positive perspective
5	Doing for	Managing negative emotions Identifying positive and negative emotions Expanding positive emotions as a strength Recognizing and enjoying the benefits of positive emotions Practicing mental control through abdominal breathing	Asking questions to restore positive emotions Asking questions about experiences that encourage confidence and happiness in daily life
6	Enabling	Enabling effective communication skills Learning assertive and responsible communication skills Analyzing communication patterns in stressful situations Finding problems and weaknesses in communication in relationships that have conflict Practicing mental control through body stabilization	Asking questions about the types of communication Asking questions on how to communicate appropriately and effectively in difficult situations and situations with conflict Applying assertive communication skills in daily life
7	Enabling	Learning a stress management strategy Finding strategies to reduce stress Finding a way to manage stress by using strengths Learning confrontation and refusing skills for unreasonable requests Learning to “butterfly embrace” Practicing mental control through abdominal breathing	Identifying stressful situations Asking questions about experiences related to using strengths to manage stress Paying compliments for overcoming stress
8	Enabling	Finding the meaning of life in everyday living Identifying changes in daily life after participating in the program Finding gratitude in daily life Finding satisfaction and meaning in life Mind control through body stabilization	Asking questions to recognize self-change in daily life through the program Asking questions about perceived meaning and happiness in life Trying to incorporate positive self-talk

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
