# Peer review of "Developing an Internet-Based Trauma Recovery Nursing Intervention Based on Swanson’s Theory of Caring for Trauma Recovery"

_ijerph, 2021, doi:10.3390/ijerph18136715_

Round 1

Reviewer 1 Report

This study presents an online program aimed at developing an Internet-based psychiatric nursing intervention that is effective in improving symptoms and restoring mental functioning in adults who have experienced post-traumatic stress disorder. The results show that mental health nurses can develop and implement an online program to care for people who have experienced trauma. In a situation where face-to-face counseling is difficult due to the Corona disaster, the possibility of developing an Internet-based counseling program is suggested, which provides an important challenge.

However, there are some modifications that should be made to the publication.

1)In the design, it is stated that the program was conducted with five non-mental health nurses as clients; it would be better to describe more about the background of the five nurses in order to implement the program.

2)In addition, the content of the program was evaluated by a specialist, and the CVI was 3.2 points, but if the five nurses who received the program as clients had evaluated the program, it would be better to include the results of the evaluation.

3)The name of the journal in citation 34, "PLoS One", not "PlOS One", to be exact.

Reviewer 2 Report

The paper is well written and makes good use of a solid conceptual framework, but I have some concerns about its basic premises and methodology.

  • The authors should consider defining the term “psychological trauma” in the first paragraph
  • Regarding this statement in the introduction: “The lack of trained professional therapists limits nurses’ ability to learn the treatment. (p. 1)” Since I am not a nurse, I am unclear regarding why a nurse should deliver therapy instead of a psychologist, marriage and family therapist, or clinical social worker. Since the concept of nurses delivering trauma therapy seems to be a basic premise of the paper, I think the authors should insert a new paragraph explaining the expanded roles of nurses in delivering behavioral healthcare and how patients benefit from treatment delivered by nurses.
  • Similarly, the following statement has the same issue as the above bullet point (p. 2): “the lack of competence of nurses in trauma intervention is thought to be a contributing factor towards the lack of participation by nurses.” Perhaps they’re not interested because it’s not within their usual scope of practice to deliver psychotherapy, and they already bear a heavy workload? Why do nurses need to deliver trauma therapy as opposed to “social welfare and clinical psychology majors?” More explanation is needed.
  • (p. 3) “The pilot test was performed using five nurses who worked in non-mental health-related areas and participated as clients. They provided feedback on their experience of the program. The researchers conducted the program in the role of the therapist.” Did these nurses report a history of trauma? Were there any inclusion/exclusion criteria besides being a nurse? Please clarify.
  • This is a premature stage for the authors’ conclusions and interpretations of the nurse participants’ feedback. The dissemination of a novel therapy usually occurs after some piloting with the target audience, e.g., patients with trauma. I do not have faith in their findings given the early stage. It might be more appropriate for the authors to write this paper as a protocol (e.g., describing the components of the intended intervention, with no results/conclusions), then follow up with a separate paper once they have data and feedback from non-nurse clients.

Round 2

Reviewer 2 Report

The removal of the term “pilot” from the abstract is an improvement. The authors’ expanded explanation of how nurses contribute to trauma care is appreciated, and adds more context to the study aim. The authors’ application of Swanson’s theory is a study strength. Please see the following critiques:

  • The work is referred to as a “pilot test” participants in the methods section, in contrast with point 5 of the editor response letter.
  • It is helpful that the authors provided more information on the nurses’ background. While I think this information should remain, it also confirms my previous view that the role of the nurses is unclear. The authors state that the clients were nurses with more than three years of nursing experience, and note their expertise in the editor response letter: “All of the nurses who participated in the evaluation worked in hospitals and had more than three years of nursing experience.”

This expertise is relevant for the purposes of intervention development. However, the target audience for this intervention is not nurses with clinical experience; the intervention is designed to treat people with trauma. There is no indication of whether the nurses had history of trauma in the editor response letter or in the manuscript. Therefore, I still struggle with authors’ decision to describe these participants as clients and describe themselves as the participants’ therapists. The authors could consider describing the participants as a convenience sample of nurses (or whatever terms they find fitting) who role played as clients for the purposes of developing this intervention.
